# Estimating an EQ-5D-3L Value Set for Romania Using Time Trade-Off

**DOI:** 10.3390/ijerph18147415

**Published:** 2021-07-11

**Authors:** Marian Sorin Paveliu, Elena Olariu, Raluca Caplescu, Yemi Oluboyede, Ileana-Gabriela Niculescu-Aron, Simona Ernu, Luke Vale

**Affiliations:** 1Romanian Academic Society, 020071 Bucharest, Romania; simona.ernu@sar.org.ro; 2Department of Pharmacology, Titu Maiorescu University, 031593 Bucharest, Romania; 3Population Health Sciences Institute, Newcastle University, Newcastle upon Tyne NE2 4AX, UK; elena.olariu@newcastle.ac.uk (E.O.); yemi.oluboyede@newcastle.ac.uk (Y.O.); luke.vale@newcastle.ac.uk (L.V.); 4Department of Statistics and Econometrics, Bucharest University of Economic Studies, 010374 Bucharest, Romania; raluca.caplescu@gmail.com (R.C.); gabriela.niculescu@csie.ase.ro (I.-G.N.-A.)

**Keywords:** cost-utility analysis, EQ-5D, utilities, health-related quality of life, quality-adjusted life years, time trade-off

## Abstract

Objective: To provide health-related quality of life (HRQoL) data to support health technology assessment (HTA) and reimbursement decisions in Romania, by developing a country-specific value set for the EQ-5D-3L questionnaire. Methods: We used the cTTO method to elicit health state values using a computer-assisted personal interviewing approach. Interviews were standardized following the most recent version of the EQ-VT protocol developed by the EuroQoL Foundation. Thirty EQ-5D-3L health states were randomly assigned to respondents in blocks of three. Econometric modeling was used to estimate values for all 243 states described by the EQ-5D-3L. Results: Data from 1556 non-institutionalized adults aged 18 years and older, selected from a national representative sample, were used to build the value set. All tested models were logically consistent; the final model chosen to generate the value set was an interval regression model. The predicted EQ-5D-3L values ranged from 0.969 to 0.399, and the relative importance of EQ-5D-3L dimensions was in the following order: mobility, pain/discomfort, self-care, anxiety/depression, and usual activities. Conclusions: These results can support reimbursement decisions and allow regional cross-country comparisons between health technologies. This study lays a stepping stone in the development of a health technology assessment process more driven by locally relevant data in Romania.

## 1. Introduction

In the current Romanian health technology assessment (HTA) process, decisions on reimbursement for the use of new technologies have not been conditioned by a threshold of effectiveness or an analysis of the budgetary impact [1]. More exactly, the process has so far used a scorecard system called “de facto” or “rapid” HTA [2,3]. This system is based, among others, on determining the number of countries where reimbursements for the use of new technologies have already been implemented, with a key role in deciding what new technologies will receive funding based on reimbursement decisions in the UK, Scotland, Germany, and France [4]. The Romanian authorities have expressed their intention to make the transition to a complete HTA process based, interalia, on cost-utility studies, using real-world data that require country-specific costs and utilities [4,5].

The determination of the costs depends on the particularities and the specific structure of the Romanian health system. On the other hand, utilities (index values) reflect the preference of the general population for different health states and are obtained using various methods, such as time trade-off (TTO), standard gamble (SG), and visual analogue scale (VAS), derived from the national general population samples [6]. The collection of utilities for different health states, also known as value sets, allows comparisons between different types of interventions and treatments for different diseases. These comparisons are essential for making decisions on how to distribute healthcare resources, thus supporting the HTA process.

The best known tool for measuring health is the EQ-5D-3L introduced by EuroQoL in the 1990s. This is an easy to administer generic tool, which allows the measurement of various health conditions during the evolution of a patient’s disease as well as the comparison of results with other disease areas. [7,8] The EQ-5D-3L consists of the descriptive system and the visual analogue scale (EQ-VAS). The descriptive system captures five dimensions of health-related quality of life (HRQoL), namely, mobility, self-care, usual activities, pain/discomfort, and anxiety/depression. EuroQol’s first instrument, EQ-5D-3L, uses only three levels of discrimination for each dimension (no problems, some problems, extreme problems). The EQ-5D-3L describes 243 health states that result from combining the three response possibilities on five dimensions. The EQ-VAS consists of a visual scale that ranges from the best condition you can imagine to the worst imaginable state, divided into 100 units. VAS is used as a quantitative measure of the perception of one’s health [6]. Later, in 2009, EuroQoL developed a tool based on five levels of discrimination and a tool dedicated to children and adolescents [9].

To measure health outcomes, country-specific index values (utilities) have been developed in many other countries: in Belgium [10], Denmark [11], France [12], Germany [13,14], Greece [15], Holland [16], Italy [17], Poland [18,19], Portugal [20], Spain [21], Slovenia [22], Sweden [23], and the United Kingdom [24,25]. The transition to the complete HTA also implies the existence of a set of index values for EQ-5D-3L based on the social preferences of the general Romanian population [20]. We aimed, as the main objective of our study, to determine the values for the different health states of EQ-5D-3L using a TTO method.

Evidence shows that there are minor differences in the value sets of countries with a comparable economic level [26,27]. These differences seem to be due to the socio-economic context, the characteristics of the health status of the population, the socio-demographic characteristics, and less due to the technique of estimation or methodology. Hence, comparing the results of cost-utility studies from countries whose value sets are very different can give misleading results and subsequently determine misuse of healthcare resources [28,29,30,31,32,33,34,35]. The premise that what is cost-effective in the UK, Germany, Scotland, or France is just as cost-effective in Romania is questionable, because Romania allocates the lowest amount for health in the EU (Romania spends only 814 euros per head per capita, 3 times less than the average for European countries) [32]. One of the main barriers to the development of HTA in Romania is precisely the absence of a standardized preference value for the EQ-5D or another HRQol instrument for measuring health outcomes. Prior to the present study, in the absence of a national set of values for any of the EQ-5D instruments, Romanian researchers have used the value set from another jurisdiction, more specifically the value set from the UK [33,34,35,36].

## 2. Materials and Methods

### 2.1. Survey Design and Sampling Framework

We conducted a cross-sectional survey in non-institutionalized persons older than 18 years, selected from the Romanian population. The data collection occurred from November 2018 to November 2019. We selected study participants from both rural and urban areas, out of a total number of 3181 settlements. Settlements were randomized and stratified to ensure representativeness, using the Romanian electoral register as the sampling frame. A total of 32 settlements (primary sampling units) were selected. Within each settlement, households and individuals were selected using a random route sampling method and the next birthday rule (the person whose birthday was closest to the interview date was interviewed), respectively [37]. Respondents signed a written consent to take part in the research and were offered no incentives for their participation. Ten percent of respondents were contacted by phone to verify that all data collection procedures were implemented as intended. 

The sample size needed for national representativeness was estimated at 1794 with a maximum error of ±3% for a confidence level of 95% and including a 10% nonresponse rate [38]. The sample size needed for an EQ-5D-3L valuation study is 300 people [16]. To meet the representativeness criteria and to allow implementation of a parallel EQ-5D-5L valuation study, the final sample size was set at 1794. This article only targets the results of the EQ-5D-3L. 

### 2.2. Interview Procedure

Interviews were face-to-face, computer-assisted, and took place in respondents’ homes. Interviewers were selected from members of patient associations or independent expert evaluators of the quality of medical services. All interviewers were trained in two 2-day training sessions conducted by the principal investigators and team members who were experts in statistics and data collection (October 2018, June 2019). One interviewer was excluded for protocol non-compliance and another five dropped out of the interviewers’ team after having performed less than 20 interviews.

The Romanian version of the EQ-5D instrument and the EQVT interviewing software were based on the English version of the EQVT software developed by the EuroQoL Foundation. The software was translated by a professional translation company.

Respondents valued three EQ-5D-3L states that were inserted at the end of the EQ-5D-5L sections (valuation tasks and EQ-5D-5L questionnaire). They also filled in the EQ-5D-3L questionnaire and self-reported their health on the EQ-VAS. Finally, the interview ended with an array of socio-demographic questions.

More details on the study’s protocol can be found elsewhere [38].

### 2.3. Valuation Protocol and Procedure

We elicited the population’s preferences using composite TTO (cTTO) techniques. The cTTO task was implemented following the most up-to-date version of the EQ-VT protocol (version 2.1) [39]. The smallest trading unit was 6 months (which by transformation means 0.05).

Values for states better than dead were elicited using the conventional TTO approach (10-year time frame). In this case, the value of a health state was defined as the number of years spent in full health considered equivalent to the number of years spent in an impaired health state divided by 10.

Values for health states worse than dead were elicited using the lead time TTO (10 years in full health followed by 10 years in the state to be valued), and were calculated as the difference between the number of years selected in the lead-time as the respondent’s indifference point and 10 divided by 10.

A quality control check developed by the EuroQoL Foundation and set up for the EQ-5D-5L valuation criteria [40] was performed weekly by the principal investigators and with the bi-monthly support of EuroQol experts. This quality control check identified interviews of suspect quality. Suspect quality was defined based on the criteria set for the EQ-5D-5L valuation study: too short explanations for the training part of the survey, not showing the worse than dead example in the training part of the survey, too short duration of the cTTO task, and not assigning the lowest value to the worse health state [40]. Based on the results of the quality control check, interviewers were sent feedback about their performance either by email or telephone.

### 2.4. Selection of EQ-5D-3L Health States

The cTTO values were collected for 30 out of the 243 EQ-5D-3L health states. The set of health states included 18 states that were selected using an orthogonal design [41], all 5 mild states (11112, 11121, 11211, 12111, 21111), and 7 other health states that had been handpicked. Ten blocks of 3 states were created and randomly allocated to participants, thus meeting the minimum number of valuations needed to develop a value set for EQ-5D-3L [16]; the same protocol has been used and published recently by other authors [42,43].

### 2.5. Statistical Analyses

#### 2.5.1. Exclusion Criteria

The following interviews were excluded:Interviews of suspect quality, performed by interviewers that were excluded from the team of interviewers due to protocol noncompliance.Interviews performed by interviewers having more than 40% of the interviews performed flagged as interviews of suspect quality (as defined in the EQ-5D-5L valuation study).Interviews performed by interviewers not performing enough interviews to achieve a harmonized learning effect between interviewers [44]. The minimum number of interviews was set at 20.Interviews for which the interviewer had not shown the worse than dead example in the training part of the survey.Participants with a positive slope on the regression between their values and the misery index of the health states assessed for participants who gave the same value to all health states or did not trade time (non-traders).

Based on these criteria, 2 datasets were created, from the most inclusive to the least inclusive for sensitivity analysis purposes:Set 1 (denoted V1) contained all valid responses and corresponded to the exclusion criterion 1.Set 2 (denoted V3) corresponded to the exclusion criteria.1, 2, 3, 4, 5.

Logical inconsistencies were identified, but respondents were not excluded from the sample based on this criterion. 

#### 2.5.2. Modeling

Based on standard practice and the following EuroQoL guidance, we tested the following models:Ordinary least squares (OLS)Robust OLSModels that account for the panel structure of the data (random intercept models with respondent and interviewer mixed effects; random coefficient models)Models that account for the censored nature of the data (tobit and interval regression models)

To estimate the value set for EQ-5D-3L, various types of linear regression models were tested. We initially considered a multiple linear regression model. Given that we expected to find patterns of heteroskedasticity in the data, we also considered a robust ordinary least squares model. This model corrects for heteroskedasticity but does not account for data clustering. Given that the analysis was performed at a health state level, but a respondent evaluated more than one health state, the data collected for various health states from the same respondent were most likely correlated. Furthermore, when collecting data with the help of interviewers, it was impossible to eliminate completely the interviewer effect. To consider these aspects in the analysis, we also developed 3 more models that account for the panel structure of the data: a model with a random intercept for respondent effect on the data, one with a random effect for the interviewer effect, and finally, a random coefficient model with random effects at the respondent level. Finally, given the censored nature of the data (values below −1 could not be assigned to health states considered worse than death), we also tested tobit and interval regression models. 

The dependent variable was computed as 1 minus cTTO (representing the loss of utility associated with the health state), and the independent variables were dummy variables created based on individual responses. A set of 10 dummy variables (MO2, MO3, SC2, SC3, UA2, UA3, PD2, PD3, AD2, AD3) was created to be used as main effects within every dimension to EQ-5D-3L [45]. Each variable represented the impact on the dependent variable generated by transitioning from level 1 (no problems) to level 2 (moderate problems), and respectively 3 (severe problems). The definitions of the variables used, as well as the model specification and methods employed, are presented in Table 1.

The final model, which will be the Romanian model, was chosen based on several selection criteria:Logical consistency and significance of parameters: All regression coefficients obtained need to be logically consistent between health states and, if possible, significant at the level of 0.05. This means that health states with severe problems on a dimension must have a lower predicted utility than health states with moderate or no problems. This holds true only for intensity levels of the same dimension. More precisely, we expected health state 11233 to be considered better than the state 11333, but we could not compare it with 21333, for example. This criterion was verified through the parameter estimates of the model, where the coefficients for level-3 states need to be higher than those for level-2 states, which, in turn, must be greater than zero.Theoretical considerations: As we expected the standard deviation of the observed values to increase with worsening severity of health states, models that accounted for heteroscedasticity were favored. If the percentage of observed values at −1 exceeded the normal range expected for valuation studies (2–10%), preference was given to censored models. For models meeting multiple criteria, preference was given to the model with the lowest number of independent variables (principle of parsimony).The goodness of fit. For all logically consistent models, we calculated the Akaike information criterion (AIC) and Bayesian information criterion (BIC). The smaller the values for AIC and BIC, the better the goodness of fit of the model. In case of different conclusions for the two indicators, BIC was preferred to also account for model parsimony.Prediction accuracy (Spearman’s correlation between predicted and observed utilities), value range, and the ranking of dimensions based on the size of the coefficient for the worst level on each dimension were also taken into account.

Therefore, we considered as candidate models for our EQ-5D-3L value set only those models whose parameters were all logically consistent and significant at the level of 0.05, which accounted for the heteroscedasticity and/or for the censored nature of the data. From the candidate models, the final model was chosen based on the AIC/BIC ranking, higher predictive accuracy, higher value range, and ranking of dimensions. 

For the final value set, the intercept was constrained to be equal to 1 (full health) if it were insignificant at the level of 0.05.

All analyses were performed on the most restrictive dataset (V3).

To determine the impact of different exclusion criteria on our final model (“the Romanian model”), we estimated and tested the model using dataset V1, which included all available interviews. Finally, if our sample was found to be very different from the general population in terms of age, sex, and place of residence, we also tested the impact of these variables on the estimated values by adjusting the model with these variables and by including survey weights in the model. Survey weights were calculated as the product of design weights (the inverse of the respondents’ probability of selection for each of the stages of the survey), nonresponse weights (percentage of people responding to the survey in each settlement), and poststratification weights. Post-stratification weights were computed using age, gender, and place of residence as variables to create poststrata. Place of residence (urban/rural) rather than the type of settlement was used to create poststrata so that each poststratum included at least 5 observations to ensure efficiency in poststratification. Population control totals for each poststratum were taken from the 2011 Romanian census.

#### 2.5.3. Comparison with Other Countries’ Value Sets

We compared our observed cTTO values with the UK values for the 14 health states that were common to both studies. We determined the observed means (Mean) and standard deviations (SD) for both countries and the predicted values (Predicted). We also tested the statistical significance of the differences between the observed means for Romania and the UK.

Finally, we compared our value set (predicted values) with the UK value set and Polish value set using a kernel distribution plot to observe the range of values, modality, or skewness.

For the entire data analysis, the significance level used was α = 0.05. The results were generated using STATA version 16 and IBM SPSS Statistics 25.

## 3. Results

A total of 1674 people were interviewed. Refusal rates varied from 0% to 73%, being higher in urban areas. The study was stopped when the minimum valid number of interviews was reached.

Of the total interviews performed, 25 respondents were excluded for having been interviewed by interviewers that were later on excluded from the team of interviewers due to noncompliance and poor interviewing performance (dataset V1: 1649 respondents). Another 81 people were excluded based on exclusion criteria b, c, and d; they had been interviewed by interviewers who had more than 40% of the interviews flagged or conducted less than 20 interviews, or the interviewer did not show the worse than dead element of the training part of the survey and no negative values were elicited for all health states presented. Finally, 12 people were excluded because they were marked either as illogical or nontraders, or had the same value (different from 1) for all evaluated states.

Only nine respondents had inconsistencies in their health state valuations in the V3 dataset, and 15 in the V1 dataset.

Sociodemographic characteristics for the final dataset (V3 = 1556), weighted and unweighted, are presented in Table 2. As shown in Table 2, women and urban areas were overrepresented in our sample. Sociodemographic characteristics for dataset V1 used for the sensitivity analysis are presented in Appendix A. The mean age was 48.50 years (SD = 16.21) for the final dataset and 48.43 (SD = 16.35) for the dataset used for sensitivity analysis (V1). The mean VAS3L was 83.45 (SD = 14.38) and the mean utility for observed health states was 0.50 (SD = 0.46) for the final dataset. Similar values were obtained for the dataset used for sensitivity analysis (V1: mean VAS3L = 83.50 (SD = 14.49); mean utility = 0.51 (SD = 0.46)).

We computed the mean, standard deviation (SD), median, and quartiles for the observed cTTO values (Table 3). The mean values ranged from 0.942 for state 21111 to −0.510 for state 33333, with similar median values (from 0.95 for states 11112, 11121, 12111, 11211, 21111 to −0.60 for state 33333). The standard deviations seemed to increase as profiles indicated worse health states, which was an early indication of heteroskedasticity. This finding is similar to those in other countries [12,16,45]. One-third of the states had no negative value evaluation (22222, 22121, 12212, 11122, 21211, 11112, 11121, 12111, 11211, and 21111), and among the rest of the sample, the percent ranged between 0.64% for state 12222 to 78.57% for state 33333. Each of the 30 states was evaluated by at least 149 respondents (Table 3).

We began our model testing process with the simplest one, the ordinary least squares model (OLS). A list of all models that were tested can be found in Appendix A. After having estimated the OLS model, we found an indication of strong heteroskedasticity in the data, which we confirmed using the Breusch–Pagan test (*p* < 0.0001). Hence, we decided that all our candidate models had to account for heteroskedasticity besides the significance and logical consistency of parameters. Table 4 presents a list of the candidate models with the highest number of consistent and significant parameters corrected for heteroskedasticity and/or accounting for the censored nature of the data. As seen in Table 4, our candidate models for the final value set were the robust ordinary least square model (ROLS), interval regression model (IRM), and interval regression model censored at −1 (IRMC). We tested all models for goodness-of-fit, focusing on the ones with the smallest AIC/BIC (Table 4). The IRM and IRMC models had the lowest AIC/BIC. Given that the prediction accuracy, the range of values, and ranking of dimensions were very similar for both IRM and IRMC, we chose IRM as our final model given its lowest AIC/BIC. The full model can be found in Appendix A.

The form of the final chosen model (IRM from Table 4), is presented below:
$$\begin{array}{cc} \mathrm{Utility}=0.032 & -\;(0.038*\text{MO2}+0.394*\text{MO3}+0.04*\text{SC2}+0.206*\text{SC3}+0.044*\text{UA2}\\ & +\;0.189*\text{UA3}+0.072*\text{PD2}+0.371*\text{PD3}+0.054*\text{AD2}+0.206*\text{AD3} \end{array}$$


ROLS, robust ordinary least-squares; IRM, interval regression model; IRMC, interval regression model censored at −1.

MO—Mobility; SC—Self-care; UA—Usual activities; PD—Pain/discomfort; AD—Anxiety/depression.

All coefficients of the dummy variables were significant at 0.05 level, meaning that having any type of problem with mobility, self-care, usual activities, pain/discomfort, or anxiety/depression significantly decreased the utility (Table 4). Predicted values for EQ-5D-3L are shown in Appendix A.

Most utility decrease was estimated for severe problems, with a cumulative impact of 1.37 utility units. This led to a negative utility of 0.4 for the health state 33333, which was the worst possible state, with more severe problems for all dimensions. Issues with mobility and pain/discomfort had the biggest impact, causing a drop in utility of 0.39 and 0.37 units, respectively. The cumulative effect of the other three dimensions was smaller than the effect of the first two taken together, which suggests that for severe problems with mobility and pain, the quality of life of a person is worse than for severe problems with anxiety/depression, being unable to take care of oneself, and carrying on with usual activities.

Moderate problems on the five dimensions had a total impact of 0.25 utility units, leading to a utility for the 22,222 health state of 0.72 units. About half of the impact came from moderate pain/discomfort (0.07) and anxiety/depression (0.05). In contrast to severe problems, in this category mobility had the smallest impact.

Based on the results obtained for the two categories, namely severe and moderate problems, the conclusion was that pain and discomfort is an important factor in perceived utility, regardless of its severity. For the other dimensions, mobility is perceived as a major impediment only if the problems are severe, while depression and anxiety matter more for moderate problems. Being able to perform self-care tasks and usual activities, while having a statistically significant impact, are not seen as major contributors to final utility.

To test the robustness of our model, we estimated and tested the IRM model (RO model) using all available responses (dataset V1) and a weighted version of dataset V3 (Table 5). The RO model performed the worst in all categories except for prediction accuracy when it was run using all available data (V1). The model performed the best when it was estimated on V3 in terms of AIC/BIC and prediction accuracy, and had similar performance in both V3 and weighted V3 datasets in terms of ranking of dimensions and number of WTD health states. The full model runs on both V1 and weighted V3 can be found in Appendix A.

Finally, we tested the prediction accuracy of the RO model in dataset V1 and the weighted V3 dataset by comparing the predicted values with the observed mean TTO values for each evaluated health state. As shown in Figure 1, the model estimated well the mean observed values in all cases.

We compared the observed cTTO values from our study with the observed TTO values from the UK MVH study [43] for the 14 health states that were common to both studies. Table 6 shows the observed means (Observed) and standard deviations (SD) for both countries, the number of respondents (n) who evaluated the health states in each study, and the predicted values (Predicted). Additionally, we tested the statistical significance of the differences between the observed means for Romania and the observed means in the UK. We found significant results at the 0.05 level for all compared states, except 33333. Values for Romania were generally higher than those recorded in the UK for all states, except 33232, for which Romanian values were significantly smaller (Table 6). Differences between health states ranged from −0.42 (for 21133) to −0.06 (corresponding to 11121) (see Figure 2).

When comparing the estimated values for all health states, the values for the Romanian EQ-5D-3L value set were higher than the values for the UK value set, but fairly similar to the values for the Polish EQ-5D-3L value set, although the estimations of individual health states differed (Figure 3).

## 4. Discussion

Our study estimated for the first time in Romania a value set for the EQ-5D-3L questionnaire. This constitutes a stepping stone to further development of HTA in Romania, as it will potentially lead to more transparent and consistent decision-making in healthcare and more efficient use of relatively scarce local resources. 

To develop our EQ-5D-3L value set, we tested several regression models. We chose the interval regression model as our final model because of all candidate models, it performed the best in terms of AIC/BIC and had similar performance with the second-best model in terms of prediction accuracy, range of values, and number of WTD health states. Our final model accounted for heteroskedasticity and all coefficients were significant at the level of 0.05. Finally, the model provided utility estimates with a range similar to the observed ones.

We compared our value set with those of the UK and Poland. We chose the UK because HTA results from the UK are often used as a guide for the Romanian HTA and because local researchers have used this value set in the absence of a local one. Even though differences were found between the two value sets, these might also be because the EQ-5D-3L valuation methodology has changed in the meantime with the use of cTTO and computer-assisted interviews. This will more likely lead to a decrease in interviewer bias, processing errors, and easier randomization of the question order. [46] We also compared our value set with that of Poland due to the higher similarities in economic and historical background with Romania. Nevertheless, intercountry differences were still observed, thus stressing the importance of using country-specific value sets for instruments such as the EQ-5D and calling for an urgent refinement of current HTA practices in Romania. This is supported by an increasing body of literature that shows that using multinational value sets or other countries’ value sets might misrepresent the value sets of individual countries [47,48].

Our sensitivity analyses performed using dataset V1 were conducted on more relaxed criteria than the primary analysis and showed that modeling can be severely undermined by data of poor quality. This is in line with other studies’ results that show that data not meeting the minimum quality criteria as set by the EQ-VT software can lead to low face validity, difficulties in data modeling, and measurement errors with a final value set not discriminating very well between more severe health states [14,49]. In our sensitivity analysis, we did not explore the effect of excluding inconsistent respondents from our model. We based our decision on the results of a systematic review of exclusion criteria in national health state valuation studies that showed that the effect of excluding inconsistent respondents on national tariffs was not consistent [50].

## 5. Limitations

### Our Study Has a Certain Number of Limitations

First of all, our study sample differed from the Romanian general population in terms of age, gender, and rural/urban distribution. We, however, corrected this imbalance by using survey weights and assessing the impact of their use on our final model and found no significant differences between weighted and unweighted analyses. Nevertheless, our survey weights did not account for other observed differences between our sample and the general population, such as education or income [51], on the values obtained. Additionally, our survey weights were based on the 2011 census data, the most recent census data available. Since 2011, migration rates have been increasing in Romania, with the country currently having the highest growth in the size of its diaspora population after Syria. Hence, our survey weights might not always have correctly adjusted the representativeness of our sample.

Third, the quality of our data would have been better had we been able to collect them using fewer well-trained interviewers. Optimally, the research should have been completed by an estimated number of 15–16 interviewers so that each would perform at least 100 interviews, thus having the time to acquire and maintain the skills of conducting interviews. Fieldwork difficulties such as the start of the data collection in winter, problems accessing certain rural areas, respondents’ reluctance to participate in the study especially in urban areas, and the large number of interviewees assigned to each interviewer led to interviewer fatigue and demotivation. Hence, the performance of our interviewers varied greatly during the study period, with several of the initial interviewers being replaced during data collection or dropping out of the interviewers’ team.

EQ-5D-3L valuation tasks were performed on the same individuals after the 10 EQ-5D-5L valuation tasks. Hence, we cannot exclude the fact that the quality of our EQ-5D-3L data might have been affected by respondent fatigue, as their attention and motivation might have dropped toward the end of the valuation task. Nevertheless, only 1.5% of the sample found the cTTO task hard to understand and only 15.9% admitted having problems deciding the value where the two lives presented were the same.

## 6. Conclusions

This is the first study conducted in Romania that estimates the index values for different health conditions in the EQ-5D-3L questionnaire, using a national population sample. These results can support reimbursement decisions and allow regional cross-country comparisons between health technologies. This study lays a stepping stone in the development of a health technology assessment process more driven by locally relevant data in Romania.

## Figures and Tables

**Figure 1 ijerph-18-07415-f001:**
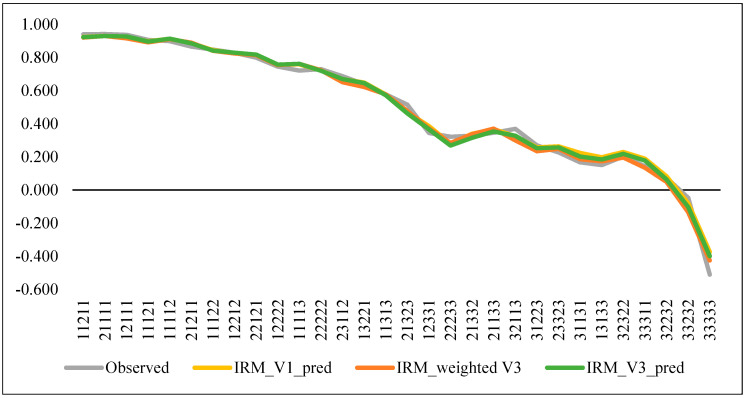
Observed and predicted values for evaluated health states.

**Figure 2 ijerph-18-07415-f002:**
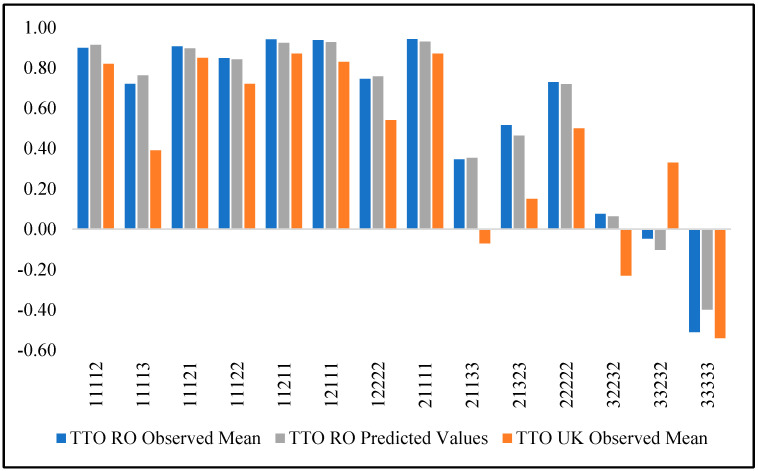
Mean observed TTO values for Romania and the UK.

**Figure 3 ijerph-18-07415-f003:**
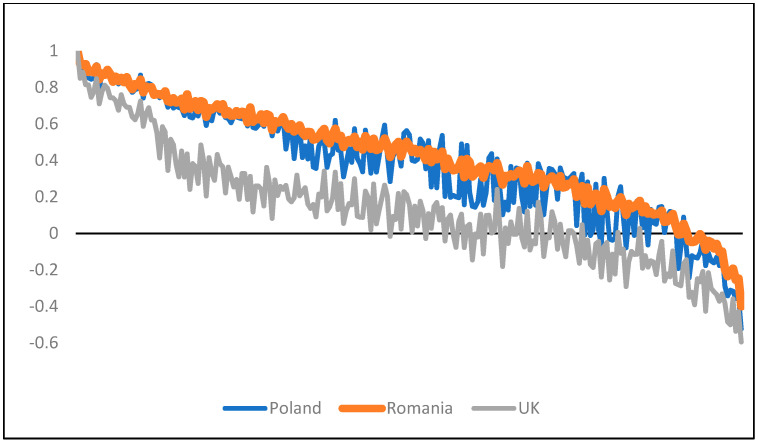
Graphical comparison of the Romanian EQ-5D-3L value set versus the UK value set and Polish value set.

**Table 1 ijerph-18-07415-t001:** Definition of variables and models.

Variable	Definition
TTO	Time Trade-Off
MO2	1 if mobility at level 2; 0 otherwise
MO3	1 if mobility at level 3; 0 otherwise
SC2	1 if self-care at level 2; 0 otherwise
SC3	1 if self-care at level 3; 0 otherwise
UA2	1 if usual activities at level 2; 0 otherwise
UA3	1 if usual activities at level 3; 0 otherwise
PD2	1 if pain/discomfort at level 2; 0 otherwise
PD3	1 if pain/discomfort at level 3; 0 otherwise
AD2	1 if anxiety/depression at level 2; 0 otherwise
AD3	1 if anxiety/depression at level 3; 0 otherwise
**Model**	**Method/f(x)**
OLS	Ordinary least squares: f(MO2,MO3,SC2,SC3,UA2,UA3,PD2,PD3,AD2,AD3)
ROLS	Robust ordinary least squares: f(MO2,MO3,SC2,SC3,UA2,UA3,PD2,PD3,AD2,AD3)
RME	Respondent-level mixed effects: f(MO2,MO3,SC2,SC3,UA2,UA3,PD2,PD3,AD2,AD3)
IME	Interviewer-level mixed effects: f(MO2,MO3,SC2,SC3,UA2,UA3,PD2,PD3,AD2,AD3)
RCM	Random coefficient model: f(MO2,MO3,SC2,SC3,UA2,UA3,PD2,PD3,AD2,AD3)
TOB	Tobit model: f(MO2,MO3,SC2,SC3,UA2,UA3,PD2,PD3,AD2,AD3)
IRM	Interval regression model: f(MO2,MO3,SC2,SC3,UA2,UA3,PD2,PD3,AD2,AD3)
IRMC	Interval regression model censored at −1: f(MO2,MO3,SC2,SC3,UA2,UA3,PD2,PD3,AD2,AD3)

**Table 2 ijerph-18-07415-t002:** Respondents’ characteristics—categorical variables, dataset V3.

Variable	Category	V3 (*n* = 1556)		Weighted V3 (*n* = 1556)		General Population
		N	%	N	%	%
MO	No problems	1288	82.8	1249	80.28	
	Some problems	264	16.97	302	19.42	N/A
	Confined to bed	4	0.26	5	0.29	
SC	No problems	1423	91.45	1386	89.07	N/A
	Some problems	129	8.29	165	10.63	
	Unable	4	0.26	5	0.29	
UA	No problems	1310	84.19	1262	81.08	
	Some problems	236	15.17	282	18.12	N/A
	Unable	10	0.64	13	0.80	
PD	No problems	1136	73.01	1106	71.07	
	Some problems	416	26.74	445	28.58	N/A
	Extreme problems	4	0.26	6	0.35	
AD	No problems	1288	82.78	1273	81.82	
	Some problems	226	14.52	244	15.70	N/A
	Extreme problems	42	2.70	39	2.48	
Gender	Female	1020	65.55	809	51.99	52%
Residence area	Urban	1139	73.20	843	54.21	55.20%
Education level	No formal education	6	0.39	11	0.72	2%
	Low	181	11.63	242	15.57	36.90%
	Medium	778	50.00	826	53.11	45.20%
	Tertiary	583	37.47	468	30.07	15.90%
	No response	8	0.51	8	0.53	
Occupation	Employed	935	60.09	829	53.28	52.10%
	Unemployed	33	2.12	50	3.23	3.90%
	Retired	386	24.81	415	26.67	26.20%
	Stay at home/domestic	107	6.88	137	8.80	7.10%
	In education	79	5.08	98	6.30	4.80%
	No response	16	1.03	27	1.72	
Income	Below the average	664	42.67	734	47.18	41.39%
	Average	271	17.42	252	16.19	30.75%
	Above the average	497	31.94	427	27.43	27.86%
	No response	124	7.97	143	9.20	

V3 Dataset corresponds to al the exclusion criteria. MO—Mobility; SC—Self-care; UA—Usual activities; PD—Pain/discomfort; AD—Anxiety/depression.

**Table 3 ijerph-18-07415-t003:** Descriptive statistics for observed cTTO values (V3 dataset).

Health State	N	Mean	SD	Median	25th Percentile	75th Percentile	Negative Values (%)
11112	162	0.899	0.106	0.95	0.85	0.95	0.00
11113	162	0.721	0.281	0.8	0.65	0.9	2.47
11121	154	0.906	0.089	0.95	0.85	0.95	0.00
11122	164	0.848	0.123	0.9	0.8	0.95	0.00
11211	155	0.941	0.061	0.95	0.9	1	0.00
11313	155	0.578	0.306	0.65	0.45	0.8	3.87
12111	151	0.937	0.069	0.95	0.9	1	0.00
12212	154	0.828	0.103	0.85	0.8	0.9	0.00
12222	157	0.745	0.151	0.75	0.675	0.85	0.64
12331	149	0.344	0.325	0.4	0.2	0.55	8.05
13133	151	0.151	0.431	0.25	0.05	0.4	23.18
13221	149	0.639	0.264	0.7	0.55	0.8	2.68
21111	152	0.942	0.058	0.95	0.9	1	0.00
21133	162	0.346	0.355	0.4	0.2	0.55	9.88
21211	158	0.866	0.099	0.9	0.8	0.95	0.00
21323	157	0.516	0.224	0.5	0.4	0.675	1.91
21332	152	0.329	0.338	0.35	0.2	0.5375	9.21
22121	149	0.800	0.117	0.8	0.75	0.9	0.00
22222	154	0.730	0.152	0.75	0.65	0.85	0.00
22233	154	0.321	0.367	0.375	0.15	0.5625	11.04
23112	152	0.688	0.209	0.7	0.6	0.8	0.66
23323	154	0.227	0.388	0.3	0.1	0.5	17.53
31131	155	0.167	0.417	0.25	0.05	0.45	20.00
31223	158	0.270	0.392	0.35	0.1	0.5	12.66
32113	164	0.369	0.346	0.4	0.25	0.6	10.98
32232	157	0.075	0.414	0.15	−0.2	0.35	28.66
32322	151	0.203	0.408	0.3	0.05	0.5	21.19
33232	164	−0.047	0.451	0.1	−0.4375	0.3	40.85
33311	158	0.143	0.479	0.25	0	0.5	24.05
33333	154	−0.510	0.421	−0.6	−0.9	−0.175	78.57

**Table 4 ijerph-18-07415-t004:** Regression analysis results.

	ROLS		IRM		IRMC	
Variable	Coefficient	SE	Coefficient	SE	Coefficient	SE
(Constant)	0.034 *	0.007	0.032 *	0.005	0.032 *	0.005
MO2	0.037 *	0.007	0.038 *	0.005	0.038 *	0.005
MO3	0.304 *	0.008	0.394 *	0.013	0.397 *	0.013
SC2	0.039 *	0.007	0.040 *	0.006	0.040 *	0.006
SC3	0.168 *	0.008	0.206 *	0.010	0.208 *	0.010
UA2	0.037 *	0.007	0.044 *	0.005	0.045 *	0.005
UA3	0.172 *	0.007	0.189 *	0.011	0.190 *	0.012
PD2	0.073 *	0.007	0.072 *	0.006	0.072 *	0.006
PD3	0.326 *	0.007	0.371 *	0.012	0.372 *	0.012
AD2	0.043 *	0.007	0.054 *	0.006	0.054 *	0.006
AD3	0.167 *	0.007	0.206 *	0.010	0.206 *	0.010
AIC	7222		−154		−21	
BIC	7297		−12		121	
Spearman’s correlation (predicted vs. observed)	0.9968		0.9954		0.9954	
Pearson’s correlation (predicted vs. observed)	0.9947		0.9959		0.9959	
U(11111)	1.000		1.000		1.000	
U(22222)	0.737		0.720		0.719	
U(33333)	−0.173		−0.399		−0.405	
No. (%) of WTD health states (%)	7 (2.9%)		22 (9%)		22 (9%)	
Mean (SD)	0.510 (0.232)		0.430 (0.282)		0.428 (0.284)	
Ranking of dimensions	PD-MO-UA-SC-AD		MO-PD-SC-AD-UA		MO-PD-SC-AD-UA	

* *p*-value < 0.05; shaded columns—final model chosen.

**Table 5 ijerph-18-07415-t005:** Sensitivity analysis results.

IRM	V3 (*n* = 1556)		Weighted V3 (*n* = 1556)		V1 (*n* = 1649)	
Variable	Coefficient	SE	Coefficient	SE	Coefficient	SE
(Constant)	0.032 *	0.005	0.039 *	0.006	0.032 *	0.005
MO2	0.038 *	0.005	0.031 *	0.007	0.037 *	0.006
MO3	0.394 *	0.013	0.416 *	0.019	0.389 *	0.013
SC2	0.040 *	0.006	0.046 *	0.007	0.044 *	0.006
SC3	0.206 *	0.010	0.228 *	0.017	0.207 *	0.010
UA2	0.044 *	0.005	0.041 *	0.006	0.043 *	0.005
UA3	0.189 *	0.011	0.183 *	0.015	0.182 *	0.011
PD2	0.072 *	0.006	0.070 *	0.007	0.069 *	0.006
PD3	0.371 *	0.012	0.359 *	0.016	0.355 *	0.012
AD2	0.054 *	0.006	0.051 *	0.007	0.054 *	0.006
AD3	0.206 *	0.010	0.200 *	0.012	0.209 *	0.010
AIC	−154		−100		62	
BIC	−12		42		205	
Spearman’s correlation (predicted vs. observed)	0.9954		0.9944		0.9963	
Pearson’s correlation (predicted vs. observed)	0.9959		0.9956		0.9962	
U(11111)	1.000		1.000		1.000	
U(22222)	0.720		0.723		0.720	
U(33333)	−0.399		−0.425		−0.374	
No. (%) of WTD health states (%)	22 (9%)		22 (9%)		19 (7.82%)	
Mean (SD)	0.430 (0.282)		0.420 (0.289)		0.438 (0.276)	
Ranking of dimensions	MO-PD-SC-AD-UA		MO-PD-SC-AD-UA		MO-PD-AD-SC-UA	

* *p*-value < 0.05; shaded columns—final model chosen.

**Table 6 ijerph-18-07415-t006:** Comparison between common evaluated states in Romania and the UK.

Health	TTO UK			TTO Romania				Difference		
State	Observed	SD	Predicted	N	Observed	SD	Predicted	Diff.	Z	*p*-Value
11112	0.82	0.29	0.85	162	0.90	0.11	0.91	−0.08	9.48	0.0000
11113	0.39	0.56	0.41	162	0.72	0.28	0.76	−0.33	14.97	0.0000
11121	0.85	0.25	0.80	154	0.91	0.09	0.90	−0.06	7.87	0.0000
11122	0.72	0.37	0.73	164	0.85	0.12	0.84	−0.13	13.36	0.0000
11211	0.87	0.23	0.88	155	0.94	0.06	0.92	−0.07	14.34	0.0000
12111	0.83	0.30	0.82	151	0.94	0.07	0.93	−0.11	19.04	0.0000
12222	0.54	0.47	0.59	157	0.75	0.15	0.76	−0.21	17.05	0.0000
21111	0.87	0.24	0.85	152	0.94	0.06	0.93	−0.07	15.44	0.0000
21133	−0.07	0.59	−0.04	162	0.35	0.35	0.35	−0.42	14.92	0.0000
21323	0.15	0.59	0.13	157	0.52	0.22	0.46	−0.37	20.46	0.0000
22222	0.50	0.47	0.52	154	0.73	0.15	0.72	−0.23	18.75	0.0000
32232	−0.23	0.57	−0.26	157	0.07	0.41	0.06	−0.30	9.22	0.0000
33232	0.33	0.51	−0.37	164	−0.05	0.45	−0.10	0.38	−10.69	0.0000
33333	−0.54	0.41	−0.59	154	−0.51	0.42	−0.40	−0.03	0.87	0.1913

## Data Availability

The data presented in this study are available on request from the corresponding author.

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
