# Peer review of "Estimating an EQ-5D-3L Value Set for Romania Using Time Trade-Off"

_ijerph, 2021, doi:10.3390/ijerph18147415_

Round 1
Reviewer 1 Report
The manuscript entitled “Estimating an EQ-5D-3L Value Set for Romania using Time Trade-Off” is well-researched work, yet I have some reservations regarding this paper’s contribution overall. I briefly explain my concerns about the current version of the paper.
-Introduction: It is really difficult to figure out the research goal of the manuscript based on the Introductory section. This section needs to be more concise and precise. In addition, it must include key information for the readers (i.e. what is a TTO? why is this the best approach in this specific case?)
-Material and Methods: The section of "Materials and Methods" must contain more information about the representativeness of the sample respect the whole population (it is not until the end of the manuscript, in the section of limitations, where the reader is informed about the use of survey weights to adjust the sample to the general population structure). In addition, it would be great to include the questions from the EQ-5D instrument. The first two paragraphs of the Results section must be moved to this section as they describe the number of discarded interviews as well as the criteria for doing so
-Results: The description of the results, mainly multivariate results, is too succinct. In addition, Tables 4 and 5 are difficult to read as we have to go through the text in the manuscript to elucidate which is the meaning of all the abbreviations
-Discussion and Conclusions: These sections can be improved by including some reflections about the impact of the results in the previous section
Author Response
Thanks for the comments and suggestions, they are objective, welcome, and have helped us increase the quality of the scientific presentation.
All were taken into account and changes were made.
1. Introduction.
Point 1: It is really difficult to figure out the research goal of the manuscript based on the Introductory section. This section needs to be more concise and precise.
Response 1:
We eliminated that information that was not absolutely necessary. We reduced the number of words and we tried to highlight the benefit that the index of values that we determined can be used in the studies that will be used for the transition to a complete hypertension process.
Point 2: In addition, it must include key information for the readers (i.e. what is a TTO? why is this the best approach in this specific case?)
Response 2:
At the first mention in the introduction, TTO is in parentheses following the notion of time trade-off. The description of the TTO technique is made in detail in the subchapter, Valuation protocol and procedure, its explanation in detail in the introduction can be redundant.
The EQ-5D-3L was developed by EuroQoL and in the process of determining the index values, they chose the TTO method. An explanation of the reasons behind its choice by EuroQoL researchers we believe goes beyond the scope of our research.
Material and Methods:
Point 3: The research design can be improved.
Response 3. The EQ-5D-3L instrument (and the following ones) have established themselves internationally precisely due to the fact that EURO QoL has made special efforts to standardize the process of valuing the sets of values specific to each country. Moreover, although EURO QoL is not directly involved in research conducted by different researchers in different countries, it imposes measures to monitor and ensure the quality of data collection and statistical analysis particularly rigorous and firm by developing applications that monitor in real-time the process and organizing teleconferences with the main investigators (in our case, the thanks in the text are addressed to the researchers who supervised and monthly guided us in our research, without being authors). In case of deviations from the quality, Euro QoL withdraws the right to use the software developed by them for valorization!! As a result, we have no way to improve the design.
Point 4: The section of "Materials and Methods" must contain more information about the representativeness of the sample respect the whole population (it is not until the end of the manuscript, in the section of limitations, where the reader is informed about the use of survey weights to adjust the sample to the general population structure).
Response 4. The limitations section is saying is that the sample collected (and for which the analysis was presented in the main text) is not representative. The authors have tried correcting the sample and compared the results of the weighted sample (not presented in the article) with the ones for the unweighted data (presented in the Results section) and they found that there are no significant differences between the two variants so they decided to present the least complex variant in order to increase reproducibility. What is more, the authors argue that the sample would have been corrected using weights based on the latest census data, which in Romania took place in 2011. The changes that occurred in the population structure in the past 10 years are important in terms of population migration and aging so even if the weights had been used, the weight-adjusted sample is still likely not representative for the population of Romania in 2019 (the year when the study took place).
Point 5: In addition, it would be great to include the questions from the EQ-5D instrument.
Response 5:
The observation is correct, the link mentioned in the bibliography being modified by EuroQoL 3 months ago. We have updated the bibliographic index in which there are also EQ-5D-3L questions.
Point 6: The first two paragraphs of the Results section must be moved to this section as they describe the number of discarded interviews as well as the criteria for doing so.
Response 6: The reviewer is right. However, for the logic of the exposure, I think it is better for these data to remain in the results, without being too much deviation from the scientific rigor, because the information is also related to the chosen models.
Results:
Point 7: The description of the results, mainly multivariate results, is too succinct. In addition, Tables 4 and 5 are difficult to read as we have to go through the text in the manuscript to elucidate which is the meaning of all the abbreviations.
Response 7: We believe Table 1 clearly explains what each abbreviation means, but we added an explanation of each abbreviation beneath the two tables in question. We extended the presentation of the results.
Discussion and Conclusions:
Point 8: These sections can be improved by including some reflections about the impact of the results in the previous section.
Response 8: Although the research required almost 2 years of work, with special and unexpected logistical difficulties and the involvement of dozens of people, in the end, the result may seem a little spectacular although it is extremely valuable, the 243 index values contained in Table S4, Predicted values for the EQ-5D-3L - Romanian model (RO model). In these conditions, the only impact, already mentioned in the text, is the removal of the last obstacle in the way of cost-utility studies specific to Romania, the existence of this set of values that will allow quantifying the answers given by patients after completing the EQ-5D- 3L and the placement of financing decisions from public funds of various medical technologies for scientific, correct reasons. After the publication of the article, all researchers who will perform such studies on Romania will have to use Table S4 (and obviously cite the source).
All changes made are highlighted in red in the attached word file, which contains the current form of the paper.

Reviewer 2 Report
The goal of this manuscript was defined.
Suggestion: Starting the literature with a global background (sentence or two) of the topic and then in Romania will grasp the reader’s attention. As seen in the manuscript this study is a first for Romania but I do suggest that adequate referencing to prior studies in the discussion should be done to support the findings of the study.
Reliability and validity of instruments were not stated. Furthermore, I do not consider myself as a language expert. However, I do suggest that this manuscript should be technical and language edited.
Author Response
Point 1. Starting the literature with a global background (sentence or two) of the topic and then in Romania will grasp the reader’s attention.
Response 1. Drawing attention to the current situation in Romania, including readers, is our intention, so we considered it appropriate to include the current situation specific to this country in the first two paragraphs and only in the following (3 and 4) the general context. Although a re-arrangement of the paragraphs is possible, we chose this order of exposition precisely to emphasize the need for our research, respectively obtaining a set of index values specific to Romania.
Point 2. As seen in the manuscript this study is a first for Romania but I do suggest that adequate referencing to prior studies in the discussion should be done to support the findings of the study.
Response 2. The last sentence in the introduction mentions all cost-effectiveness studies (cost-utility studies) using country-specific costs, but index values of UK utilities. Unfortunately, other studies of this type or other attempts to determine the index values specific to Romania do not exist.
Point 3. Reliability and validity of instruments were not stated.
Response 3. The observation is correct. We appreciate that studies on the reliability and validity of EQ-5D-3L used explicitly to assess the health of patients in Romania undergoing various treatments can be performed only after the index values that represent the result of our study will be applied. Any comments in this regard would have been strictly speculative. Of course, we could introduce at the end of the article a phrase to remind that ”future studies that will be based on EQ-5D-3L using the set of index values specific to Romania can be compared with studies from other countries, thus providing indirect clues about the reliability and validity of the instrument and/or the set of index values determined by us.”
Reviewer 3 Report
Congratulations to the authors,
The structure of the article is adequate.
The article is suitable.
The work is interesting.
The methodology of the article is adequate and interesting.
I present the following recommendations for the improvement of the article:
I advise including a bibliography of the last years 2020. Also, if the authors can include a bibliography of the year 2021, it would be very interesting.
In the bibliography, I believe that if the bibliography from number 52 to 58 belongs to the authors of the article. I think they should remove some references. I think there are too many for the same authors.
Authors must properly format the bibliography.
The conclusions section is scarce. Authors should expand this section. In addition, the authors must include future lines of research in this section.
Author Response
Thanks for the comments and suggestions, they are objective, welcome, and have helped us increase the quality of the scientific presentation.
Point 1: I advise including a bibliography of the last years 2020. Also, if the authors can include a bibliography of the year 2021, it would be very interesting.
Response 1: The study was completed in 2020 and the final result is in Table S4. Except for the bibliographic index 6, which is important because it refers to the user guide of the EQ-5D-3L instrument, which was updated by EuroQoL 3 months ago, it is inappropriate to update the bibliography, the one mentioned being the one we relied on our research.
Point 2: In the bibliography, I believe that if the bibliography from number 52 to 58 belongs to the authors of the article. I think they should remove some references. I think there are too many for the same authors.
Response 2: Thank you for reading the material so carefully! Simply, the person who transferred our article to the journal format forgot to remove the guide for formatting the bibliography. The last bibliographic index is 51. Indices 52-58 have been removed.
Point 3: Authors must properly format the bibliography.
Response 3: You are perfectly right. Out of the desire to offer the article to the editors in the form that respects the editing conventions, we will make corrections to the bibliography form only after the article will be accepted or is close to the final form. The bibliography in the unformatted word file was generated using Mendeley Cite-O-Matic, in the format of this application. Any changes made before the pre-final form will be removed at the first refresh of the bibliography and we should make corrections again. We will pay attention to this observation.
Point 4: The conclusions section is scarce. Authors should expand this section. In addition, the authors must include future lines of research in this section.
Response 4: The observation regarding the size of the conclusions and the nuance of including future lines of research that could be followed is correct. Although the research required almost 2 years of work, with special and unexpected logistical difficulties and the involvement of dozens of people, in the end, the result may seem a little spectacular although it is extremely valuable, the 243 index values contained in Table S4, Predicted values for the EQ-5D-3L - Romanian model (RO model). In these conditions, the only impact, already mentioned in the text, is the removal of the last obstacle in the way of cost-utility studies specific to Romania, the existence of this set of values that will allow quantifying the answers given by patients after completing the EQ-5D- 3L and the placement of financing decisions from public funds of various medical technologies for scientific, correct reasons. After the publication of the article, all researchers who will perform such studies on Romania will have to use Table S4 (and obviously cite the source).
In our opinion, a possible survey of how the population values different health conditions do not make sense in the near future (5-10 years) because economic conditions and cultural factors do not change so quickly as to affect the set of values. determined in this paper. From this perspective, we believe that it is more appropriate to limit ourselves to what we have managed to achieve. So far, we have not found such a study to be performed twice in the same country.
Round 2
Reviewer 1 Report
I have no further comments. The manuscript has improved enough